# Identification and Screening of Novel Antimicrobial Peptides from Medicinal Leech via Heterologous Expression in *Escherichia coli*

**DOI:** 10.3390/ijms26146903

**Published:** 2025-07-18

**Authors:** Maria Serebrennikova, Ekaterina Grafskaia, Daria Kharlampieva, Ksenia Brovina, Pavel Bobrovsky, Sabina Alieva, Valentin Manuvera, Vassili Lazarev

**Affiliations:** 1Moscow Center for Advanced Studies 20, Kulakova Str., Moscow 123592, Russia; maria.serebrennikova.msu@gmail.com (M.S.); ksenia.brovina33@yandex.ru (K.B.); pbobrovskiy@gmail.com (P.B.); lazarev@rcpcm.org (V.L.); 2Laboratory of Genetic Engineering, Lopukhin Federal Research and Clinical Center of Physical-Chemical Medicine of Federal Medical Biological Agency, Moscow 119435, Russia; harlampieva_d@mail.ru (D.K.); stadashi6@gmail.com (S.A.)

**Keywords:** antimicrobial peptides, antibacterial activity, medicinal leech, *Hirudo medicinalis*, gene expression, *Escherichia coli*, genetic engineering

## Abstract

The growing threat of infectious diseases requires novel therapeutics with different mechanisms of action. Antimicrobial peptides (AMPs), which are crucial for innate immunity, are a promising research area. The medicinal leech (*Hirudo medicinalis*) is a potential source of bioactive AMPs that are vital while interacting with microorganisms. This study aims to investigate the antimicrobial properties of peptides found in the *H. medicinalis* genome using a novel high-throughput screening method based on the expression of recombinant AMP genes in *Escherichia coli*. This approach enables the direct detection of AMP activity within cells, skipping the synthesis and purification steps, while allowing the simultaneous analysis of multiple peptides. The application of this method to the first identified candidate AMPs from *H. medicinalis* resulted in the discovery of three novel peptides: LBrHM1, NrlHM1 and NrlHM2. These peptides, which belong to the lumbricin and macin families, exhibit significant activity against *E. coli*. Two fragments of the new LBrHM1 homologue were synthesised and studied: a unique N-terminal fragment (residues 1–23) and a fragment (residues 27–55) coinciding with the active site of lumbricin I. Both fragments exhibited antimicrobial activity in a liquid medium against *Bacillus subtilis*. Notably, the N-terminal fragment lacks homologues among previously described AMPs.

## 1. Introduction

In the context of widespread antibiotic resistance, developing new therapeutic agents with unique mechanisms of action is becoming a crucial challenge in the fight against infectious diseases. In this regard, antimicrobial peptides (AMPs) have attracted particular attention. These peptides play a key role in the innate immunity of many organisms by providing the first line of defence against pathogens. These naturally occurring peptides demonstrate a wide range of antimicrobial activity against bacteria, viruses and fungi [1]. The study of antimicrobial peptide diversity in various natural sources is an important area of scientific research, aimed at identifying new molecules with unique properties and mechanisms of action.

The medicinal leech (*Hirudo medicinalis*) has a complex system for regulating its interactions with microorganisms, enabling it to closely cohabit with them. Its saliva contains many diverse molecules with a wide range of biological activity [2,3,4,5]. These include well-known substances such as the anticoagulant hirudin [6,7] and histamine, which has pro-inflammatory properties [5]. However, leech saliva contains many other molecules, including antimicrobial peptides necessary to counteract infections that enter the animal’s body through its bite [8,9,10,11,12]. The sophisticated defence mechanisms of leeches against potential pathogens emphasise their value as a source of new antimicrobial agents.

To date, several components of the medicinal leech’s innate immune system have been discovered. One such protein is destabilase [8], which has been well studied. It belongs to the I-type lysozyme family, combining muramidase, isopeptidase and non-enzymatic antibacterial activities [13,14]. Several classical antimicrobial peptides have also been described in leeches, including lumbricin I, which was first discovered in the earthworm *Lumbricus rubellus* [9] and described for the medicinal leech *Hirudo medicinalis* [10]; neuromacin, characterised for the medicinal leech *Hirudo medicinalis* [10]; theromacin and theromyzin, isolated from the coelomic fluid of the common bird leech *Theromyzon tessulatum* [11]; macins, identified in the genome of the Asian buffalo leech *Hirudinaria manillensis* [15]; or hirudomacin, found in the salivary gland of the Japanese leech *Hirudo nipponica* Whitman [12].

Despite the progress that has been made, the potential of medicinal leeches as a source of new antimicrobial peptides has barely been tapped into. Modern research methods tend to focus on analysing either large proteins or short peptide sequences. This means that medium-length peptides are insufficiently studied due to a lack of suitable tools to overcome methodological limitations, hindering the discovery and characterisation of new antimicrobial compounds. Traditional methods involving the production and testing of individual recombinant proteins are labour-intensive, expensive and, most importantly, unsuitable for studying peptides. This is why these methods are unable to analyse large numbers of peptide sequences. The chemical synthesis of long peptides is also a complex task that requires considerable effort, thereby reducing research productivity. Therefore, effective primary screening methods are needed to allow a quick and rational assessment of the antimicrobial potential of a large number of putative AMPs.

The aim of this study was to investigate the antimicrobial properties of peptides found in the genome of the medicinal leech and to determine their potential using a new primary screening method based on the expression of recombinant antimicrobial peptide genes in *Escherichia coli* [16]. The advantage of this approach is that it allows AMP activity to be directly detected in cells, bypassing the synthesis and purification steps. This simplifies the identification process for active compounds and enables the simultaneous analysis of a large number of peptides. Our previous works [17,18,19] involved thoroughly analysing the peptide composition of the medicinal leech to identify potential antimicrobial peptides. Some of these were characterised and validated using traditional assay methods. However, a significant proportion of potential AMPs consisting of more than 60 amino acids remained unexplored, as approaches that worked well for short peptides were inapplicable to longer sequences. The introduction of a new primary screening method, which overcomes these limitations, opens up new possibilities for studying the antimicrobial potential of long peptides that were previously unavailable for detailed analysis. This in turn may increase our understanding of antimicrobial activity, which is important for developing effective tools to combat infectious diseases.

## 2. Results

### 2.1. Identification of AMP Homologues and Comparative Analyses

Medicinal leech attract the attention of scientists due to their unique set of biologically active substances, including antimicrobial peptides. This study analysed the *H. medicinalis* genome (NCBI BioProject PRJNA257563) and revealed 10 sequences that are homologous to known antimicrobial peptides found in medicinal leeches (Table A1). These sequences corresponded to three families based on their homology with proteins from *L. rubellus* and *H. medicinalis*: lumbricin LBr-LR (UniProt ID O96447), neuromacin Nrm-HM (UniProt ID A8V0B3) and theromacin Thm-HM (UniProt ID A8I0L8). For the alignments shown in Figure 1, the reference sequences used in each group were LBr-LR, Nrm-HM and Thm-HM, respectively. The percentage similarities reported correspond to comparisons made against these reference sequences. Five lumbricin homologues (LBrHM1, LBrHM11, LBrHM12, LBrHM13 and LBrHM14) showed a level of similarity to the LBr-HM reference sequence ranging from 51% to 69% (Figure 1a). Two of the four neuromacin homologues (NrmHM1 and NrmHM2) differ from the Nrm-HM reference sequence by only one amino acid (98% similarity level), whereas the remaining two, NrlHM1 and NrlHM2, differ more strongly from Nrm-HM, demonstrating 68% and 64% similarity, respectively (Figure 1b). The theromacin homologue (ThmHM1) showed the lowest similarity to the Thm-HM reference sequence at 36% (Figure 1c).

Comparing the predicted sequences with known antimicrobial peptides allowed us to evaluate their conservatism and infer the functional significance of the identified sequences.

### 2.2. Sequence Analysis

To confirm the correspondence between the sequences of the potential antimicrobial peptides found in the genome and the real data, PCR was performed using cDNA as a template. The results showed a complete match between the cDNA and the genomic sequences for most of the peptides. However, two variants of LBrHM11 were identified: one of these (LBrHM11*) differed from the genomic sequence by one cytosine nucleotide residue at position 143.

Given the previous analysis outcome, plasmid vectors based on pET-22b(+) were constructed, containing the genes of the potential antimicrobial peptides that were amplified from the cDNA. Consequently, eleven experimental and five control constructs were created (Table 1).

### 2.3. Antimicrobial Activity Was Evaluated Using the Escherichia coli Expression System

After obtaining constructs containing sequences of putative AMPs, the next step was to study their antimicrobial activity. To achieve this, we used a primary screening method involving the transformation of *E. coli* bacterial cells with plasmids carrying a DNA fragment-encoding AMP under the control of an inducible promoter. First, we tested the growth of bacteria carrying AMP-containing plasmids in a liquid medium, determining the OD_600_ value every hour for eight hours (Figure 2 and Table A2). The pET-min plasmid, which was based on the pET-22b(+) plasmid but does not carry the AMP gene, was used as a negative control. The positive controls were the pET-22b(+)-LBR-LR, pET-22b(+)-NRM-HM and pET-22b(+)-THM-HM plasmids, which contain peptide sequences obtained from a homologue search. The pET-22b(+)-mel plasmid was also included as a positive control, as it encodes the sequence of the known antimicrobial peptide (AMP) melittin. The results showed that the peptides LBrHM1, NrlHM1 and NrlHM2 demonstrate antimicrobial activity comparable to melittin, while the other peptides from the lumbricin group, including the control peptide LBr-LR, are inactive.

To assess the impact of AMP expression on bacterial growth, a statistical analysis was conducted using the non-parametric Mann–Whitney U-test with a significance level set at *p*-value < 0.05. Data analysis for the first 4 h was not performed due to the lack of significant differences between the groups. Starting from the 5th hour, most of the tested samples showed significant differences compared to the negative control, with the exception of samples LBrHM12 and LBrHM13. Notably, a significantly higher antimicrobial activity was observed for the lumbricin homolog LBrHM1 compared to the positive control, whereas for the other peptides, despite the presence of significant differences with the positive control, the observed effect was less pronounced.

In the second stage, serial dilutions of bacterial cultures were spotted onto the surface of LB agar medium as an alternative to measuring optical density. Due to the nature of the test, the result is presented as a digital interpretation with colouring according to the assigned score. The modal values of the antimicrobial activity analysis based on eight biological and two technical replicates using drop serial dilution of bacterial cultures in the range from 1 to 10^4^ are shown in Figure 3a. Figure 3b shows an example of assigning a digital value according to cell growth. The same trend in activity levels as in the first stage was observed in the second stage: three leading peptides and a group of inactive lumbricin homologues, which coincided with the results of the optical density analysis. This confirms the stability of the differences identified in the antimicrobial efficacy of the peptides under study.

### 2.4. Selection of Peptide Fragments for Chemical Synthesis

The sequence of lumbricin LBrHM1 was examined in detail based on the results of the antimicrobial activity analysis. Particular focus was given to the unique N-terminal fragment LBrHM1(1–23) and the fragment LBrHM1(27–55) (Figure 4). Lumbricin LBrHM1 was selected for further study due to its unique activity profile. Unlike other lumbricin homologues that have been identified, LBrHM1 demonstrated significant antimicrobial activity in the conducted experiments. To verify these results, its activity was confirmed through studies using synthetic peptides. Previous work on the synthesis and analysis of the lumbricin I (6–34) fragment demonstrated the promise of this approach [9]. Additionally, the presence of a unique region in the LBrHM1 sequence that has no analogue in known antimicrobial peptide databases makes it an interesting subject for further research. To comprehensively study its antimicrobial properties, the complete LBrHM1 peptide and two isolated fragments were synthesised.

### 2.5. Characterisation of the Antibacterial Activity of Synthetic Peptides

To evaluate the contribution of the LBrHM1(1–23) and LBrHM1(27–55) fragments to the peptide’s overall activity, the antimicrobial activity of the full-length synthetic peptide LBrHM1 and its two fragments was analysed. The results are presented in Table 2 as minimum inhibitory concentration (MIC) values and colony-forming units (CFUs). The N-terminal fragment, LBrHM1(1–23), exhibits high antimicrobial activity against both *E. coli* and *B. subtilis*. The second fragment, LBrHM1(27–55), is mainly active against *B. subtilis*. These results suggest that different parts of the LBrHM1 peptide contribute differently to its antimicrobial properties.

## 3. Discussion

We identified several new potential antimicrobial peptides in our earlier research into the genome, transcriptome and proteome of the medicinal leech [17,18,19]. Due to their relatively short sequence, some of these peptides could be synthesised and tested. Unfortunately, a substantial number of potential antimicrobial peptides have been overlooked due to the lack of a reliable method for their evaluation. Our search strategy aimed to identify potential homologues of known peptides in order to expand our understanding of AMP diversity in *H. medicinalis*. In the process, 11 new sequences, ranging from 54 to 83 amino acids in length and presumably encoding antimicrobial peptides, were identified (Table A1). Among these, six sequences showed homology with lumbricins, four with neuromacins and one with theromacin. These peptide families were selected for primary analysis due to the limited amount of published data on AMPs found in different leech species. A complete list of peptides previously described in the literature and used as the basis for the analyses is presented in Table 3.

As a result of the bioinformatic analysis (Figure 1), homology was identified between the discovered sequences and three families of known antimicrobial peptides. Lumbricins demonstrate similar proline patterns: P-X4-P-X5-P-X11-P. LBrHM1 has a unique N-terminal fragment that is absent in other members of the family. Neuromacins and theromacins retain characteristic cysteine patterns, C-X6-C-X13,14-C-X3-C-X9-C-X6,7-C-X6,9,10-C-X-C and C-X3-C-X2-C-X7-C-X7-C-X9-C-X-C-X10,13-C, respectively, which are similar to those reported in [25].

Lumbricin I [9], the first and most widely studied peptide in the lumbricin family, exhibits antibacterial activity against a variety of bacteria, including Gram-positive species such as *Staphylococcus aureus*, *Streptococcus mutans* and *Bacillus subtilis*. It also exhibits activity against Gram-negative species like *Escherichia coli* and fungi such as *Candida albicans* and *Saccharomyces cerevisiae* [9,10,20,21,26]. Lumbricins are antimicrobial peptides (AMPs) which contain multiple proline residues and aromatic amino acids, such as phenylalanine, tyrosine and tryptophan. In the lumbricin-related sequences we identified, the proline content ranges from four to six residues (average 7.9% of sequence length), and the number of aromatic amino acid residues ranges from 7 to 11 (average 13.1% of sequence length).

So far, theromacins and neuromacins have been shown to exhibit high antimicrobial activity against Gram-positive bacteria such as *Bacillus megaterium*, *Micrococcus luteus* [23] and *Micrococcus nishinomiyaensis* [10], and lower activity against Gram-negative bacteria such as *Escherichia coli* [11]. A key aspect of their antimicrobial activity is their ability to aggregate bacteria and effectively permeabilise their membranes [27]. Macins are cationic cysteine-rich AMPs, with a highly conserved primary structure. The sequences we classified as macins contain between seven and nine cysteine residues (on average 12.7% of the sequence length). It is noteworthy that the theromacin sequence found to show the least homology with known theromacins retains similarity predominantly in regions containing cysteine residues. This fact may indicate potentially novel functional or structural features of this peptide, which requires further investigation. Information on the found peptides is presented in Table A1.

Analysis of the correspondence between cDNA and genome sequences showed their complete coincidence for most of the peptides studied. The only exception was peptide LBrHM11, for which an alternative variant (LBrHM11*) was identified that differs from the genomic sequence by one nucleotide, cytosine, at position 143. Interestingly, this nucleotide polymorphism results in the substitution of the amino acid leucine (L) for proline (P).

This study identified potentially new antimicrobial peptides. However, the analysis alone does not provide information on the actual antimicrobial activity of these peptides, hence the need for experimental verification of their biological properties. One of the key aspects to consider is the size of the identified peptides. The fact that they are much longer than the peptide fragments typically synthesised for AMP studies may make them difficult to obtain by conventional methods. In earlier studies, such peptides were isolated directly from leech tissues [10], which is a time-consuming and limiting approach. Another concept relies on the synthesis of only shortened fragments that constitute the most conserved part of the native peptide, as was performed, for example, for a peptide from an earthworm [9].

Due to these difficulties and limitations of traditional methods for the preparation and study of long peptides, an alternative approach that circumvents these obstacles has recently been actively developed. Thanks to a new but well-established method of analysing the antimicrobial activity of peptides by expression of their encoding genes [16], it is now possible to test other promising, but previously inaccessible due to their length, putative AMPs. It relies on the study of bacterial cultures transformed with pET-22b(+) based plasmids (Figure 1). The encoded AMPs contain the signal peptide PelB, which enables their transport into the periplasmic space.

The strategy we chose for the initial screening of identified AMPs allows us to rapidly assess the effect of recombinant peptide expression on bacterial cell growth. Antimicrobial activity was assessed using two complementary approaches: kinetic measurement of optical density in liquid medium and drop seeding of serial dilutions of bacterial cultures onto agar medium. Measuring the OD_600_ value enables the quantification of bacterial cell growth in the presence or absence of the AMP gene transcription inducer. Drip seeding of serial dilutions allows visual assessment of bacterial colony density, making it possible to detect even weakly expressed antimicrobial activity. The combination of these approaches increases the reliability of the results and provides a more complete picture of the antimicrobial potential of the peptides under study.

The results obtained by the analysis revealed notable differences in the activity of the tested compounds (Figure 2 and Figure 3a). Based on them, all tested peptides were divided into three groups. LBrHM1, belonging to the lumbricin family, and NrlHM1 and NrlHM2, belonging to the neuromacins, demonstrated the highest antimicrobial activity comparable to that of the well-known AMP, melittin. The remaining peptides from the macin family showed moderate antimicrobial properties. In contrast, other lumbricin homologues, including LBrHM14, identical to the LBr-HM sequence previously described for lumbricin from the medicinal leech [10], showed no antimicrobial activity under the conditions tested. It is possible that the activity of individual peptides requires specific conditions or interactions with other molecules not present in the *E. coli* model system. It is also possible that they do not possess antimicrobial activity directly against *E. coli*.

In comparison with other known control peptides used for homologue search, it is worth noting that the antimicrobial activity of full-length LBr-LR, which would be synthesised rather than directly isolated from *L. rubellus*, has not been experimentally proven before. Nevertheless, a study by Cho et al. [9] showed that its MIC is 12 μM for both *E. coli* and B. subtilis. Therefore, we can mainly rely on the data from our experiment where LBr-LR shows minimal activity under test conditions, along with other sequences of its homologues, except for LBrHM1, which shows high antimicrobial activity within our study. Similarly, limited information is available for macins, indicating only an MBC of 25 μM for neuromacin against *E. coli* [23]. At the same time, according to our results, Nrm-HM and Thm-HM show high antibacterial activity. It is important to note that under the conditions of our experiment, the peptides from the NrmHM group show comparable or even better activity, while the peptides from the NrlHM and ThmHM groups show similar or slightly lower activity compared to the positive controls Nrm-HM and Thm-HM.

Despite some disadvantages of the method used, including the inability to assess the expression level, stability and real natural specificity of the tested AMPs, it offers several advantages. These advantages lie in the possibility of testing native, full-length peptides without the need for their chemical synthesis. Furthermore, the method has a high throughput capacity, which allows analysing a large number of peptides simultaneously. As this system has a relatively low probability of producing false positive results, peptides that have demonstrated activity can be relied upon to be antimicrobial. However, it should be noted that at the same time there is a rather serious possibility of obtaining false negative results and discarding active AMPs, which is an undoubted inherent limitation of the method used.

To further investigate the antimicrobial properties of the LBrHM1 peptide, we focused on synthesising and studying individual fragments (Figure 4). Unlike the macins, which demonstrated activity in the tests, the other lumbricins did not exhibit significant antimicrobial properties, prompting further investigation of LBrHM1. The N-terminal fragment (residues 1–23) was therefore selected for investigation, as it represents a unique sequence that is not found in any of the previously described lumbricins. A second fragment (residues 27–55) was also selected for study, as it correlates with the region of lumbricin I (6–34) that has been shown to be more effective than the full-length lumbricin I described by Ju Hyun Cho et al. [9].

In liquid medium, both LBrHM1 fragments exhibit slight antimicrobial activity only at the maximum concentration of the peptide tested and only against *B. subtilis*. At the same time, when tested on agar medium, the N-terminal fragment completely inhibits the growth of both *E. coli* and *B. subtilis* cultures, whereas the second fragment is less active and only against *B. subtilis*. The full-length peptide LBrHM1 weakly inhibits the growth of both cultures (Table 2). In our opinion, the difference in the results obtained in two different tests is due to the low solubility of synthetic peptides. As already mentioned, when working with liquid cultures, we had to use a suspension of peptides rather than a true solution. At the same time, during prolonged incubation on the surface of agar medium, peptides can diffuse, which increases their effective dose.

Thus, the data obtained indicate that both fragments make a significant contribution to the antimicrobial activity of LBrHM1, with the N-terminal fragment demonstrating a broader spectrum of action and pronounced efficacy. As mentioned above, the N-terminal fragment does not have similar sequences among known AMPs, which makes it particularly interesting from the point of view of further studies and the search for its homologues.

## 4. Materials and Methods

### 4.1. Identification of New AMP Sequences

Homologues of the following peptides were searched for to determine evolutionary conservativity and the potential existence of similar sequences: lumbricin from *L. rubellus* [9], neuromacin from *H. medicinalis* [23] and theromacin from *H. medicinalis* [10]. The *H. medicinalis* genome [28] (in the NCBI database under the number PRJNA257563) was used as the basis of the search. The protein-coding sequences of the genome were analysed using the BLAST algorithm tblastn (v2.16.0, accessed on 17.07.2025) [29]. The key criterion for selection was the level of sequence homology, determined by the percentage identity calculated during the search process. The minimum value was 35%. Sequence homologues of known *H. medicinalis* AMPs were selected based on similarity to three known AMPs from different families.

### 4.2. Bacterial Medium

Bacterial medium LB: tryptone 10 g/L, yeast extract 5 g/L, NaCl 10 g/L. NaCl-reduced LB medium: tryptone 10 g/L, yeast extract 5 g/L, NaCl 5 g/L. Agar LB medium with reduced NaCl: tryptone 10 g/L, yeast extract 5 g/L, NaCl 5 g/L, 1.5% agar.

Bacterial medium MHB: Mueller–Hinton broth (BD Difco, Thermo Fisher Scientific Inc., Waltham, MA, USA) 21 g/L. MHB agar medium: Mueller–Hinton broth (BD Difco, Thermo Fisher Scientific Inc., Waltham, MA, USA) 21 g/L, 1.5% agar.

### 4.3. E. coli Strains

*E. coli* Top 10 strain (F-mcrA Δ(mrr-hsdRMS-mcrBC) φ80lacZΔM15 ΔlacX74 nupG recA1 araD139 Δ(ara-leu)7697 galE15 galK16 rpsL(Str^R^) endA1 λ^−^) (Invitrogen, Carlsbad, CA, USA) was used in the construction of recombinant plasmids. *E. coli* BL21-gold (DE3) strain was used in the antimicrobial activity assay: *E. coli E. coli* B F–ompT hsdS(rB–mB–) dcm+ Tetr gal λ(DE3) endA (Novagen, Madison, WI, USA).

### 4.4. Plasmids

The recipient vector was pET-22b(+). The plasmid pET-22b(+)-mel was used as the positive control, encoding the well-known, potent AMP melittin [30]. The negative control was the pET-min plasmid, which was based on the commercial pET-22b(+) plasmid but with the region that encodes the signal peptide removed. Using a negative control enables the influence of the plasmid and the transformation process on bacterial growth to be excluded. Using a positive control allows the effectiveness of the method to be verified and enables the activity of the peptides under study to be evaluated regarding a known antimicrobial agent.

### 4.5. Plasmid Construction and Expression of Recombinant AMPs

The selected peptides were obtained from *E. coli* cells using the commercial plasmid pET-22b(+) (Novagen, Madison, WI, USA). The process of construction was carried out using the polymerase incomplete primer extension (PIPE) method [31]. Plasmids encoding the candidate polypeptides were obtained through several steps. First, full-length PCR amplification of the pET-22b(+) vector was performed using PETPIPE-2N and 22b Rp primers (Table A3, № 1–2). Second, PCR amplification of the coding regions was carried out using specific primers (Table A3, № 3–19). A cDNA library of *H. medicinalis* salivary gland and muscle tissue cells served as the template for this second PCR amplification. The purified PCR fragments and the vector were then mixed in a 5:1 molar ratio. The reaction products were then transformed directly into *E. coli* Top10 cells and seeded onto Petri dishes containing ampicillin (150 μg/mL) in an agar LB medium. The resulting colonies were analysed by PCR of the cell suspension using T7 and T7t oligonucleotides (Novagen, Madison, WI, USA) (Table A3, № 20–21). Colonies containing plasmids with inserts were transferred to a liquid LB medium. Plasmid DNA was then isolated from the culture by alkaline lysis. The structure of the obtained plasmids was confirmed by Sanger sequencing using an AbiPrism 3730xl capillary sequencer (Applied Biosystems, Foster City, CA, USA).

As the cloning steps are the same for all selected sequences encoding antimicrobial peptides, a schematic fragment of the plasmid map designed for expressing the control antimicrobial peptide melittin is presented for clarity (Figure 5).

### 4.6. Testing Antimicrobial Activity by Analysis Using the Escherichia coli Expression System

The antimicrobial activity was evaluated using *E. coli* BL21-gold(DE3) cells that had been transformed with the pET-22b(+) series of plasmids, which encode the corresponding peptides. This process was carried out as described in the paper [16]. Two technical replicates were performed. Both liquid and solid LB medium with reduced NaCl content were used for the tests.

Following transformation, the cells were seeded onto agar plates containing ampicillin (150 μg/mL) and glucose (0.5 g/L) and incubated at 30 °C for 19–20 h.

Single colonies were scraped into the wells of sterile 96-well plates (Eppendorf, Hamburg, Germany) containing LB medium (ampicillin 150 μg/mL) and incubated for one hour at 37 °C in an MB100-2A thermoshaker (ALLSHENG, Hangzhou, China), in accordance with the methodology [16]. Eight colonies were selected for each sample and positive control and six for the negative control. Several wells containing pure medium were used as sterility controls.

To evaluate the antimicrobial activity in a liquid medium, the bacterial culture was transferred to LB medium with a 10-fold dilution of the culture, with or without IPTG (0.1 mM), and incubated in a thermoshaker. The optical density of the culture was measured every hour at a wavelength of 600 nm (OD_600_) using a Microplate Reader AMR-100 tablet photometer (ALLSHENG, Hangzhou, China). A total of eight measurements were taken.

To evaluate the antimicrobial activity in an agar medium, the split clones were incubated in LB medium (containing ampicillin at a concentration of 150 μg/mL) for one hour at 37 °C in a thermoshaker. A series of dilutions were then performed (twofold and tenfold, ranging from 10^1^ to 10^4^). For each clone, an initial suspension and each dilution were made, and a 5 μL drop of each was transferred to the surface of LB agar medium (containing ampicillin at 150 μg/mL) with or without IPTG (0.1 mM). After the drops had been completely absorbed and dried, the plates were incubated at 37 °C for 22 h, after which the growth or absence of cells was visually assessed.

### 4.7. Statistical Analysis

Statistical analysis of the data was performed using Python software (v3.11.13). The non-parametric U-test, as implemented in the SciPy module “https://docs.scipy.org/doc/scipy/reference/generated/scipy.stats.mannwhitneyu.html” (v1.16.0, accessed on 17 July 2025), was utilised for the analysis. Differences were considered significant at a *p*-value < 0.05.

### 4.8. Chemical Synthesis and Purification of Peptides

The peptides were prepared by the solid-phase method using the N-9-fluorenylmethyloxycarbonyl (FMOC) strategy on a Liberty Blue automated microwave peptide synthesizer (CEM, Stallings, NC, USA) [32].

The peptides were then purified using a Zorbax SB-C18 column (Agilent, Santa Clara, CA, USA) on an AKTA Pure 25 M1 chromatograph with UV detection (GE Healthcare Life Sciences, Malbourough, MA, USA) in reversed-phase HPLC mode. The fractions corresponding to the major peaks on the chromatogram at the elution stage were collected manually.

Once purification was complete, an HPLC system with Shimadzu LCMS-2020 mass spectrometric detection (Shimadzu, Kyoto, Japan) was used to confirm the qualitative composition of all collected fractions. Based on the results of the mass spectrometric analysis, conclusions were made about the composition of the chromatographic peaks, and the fractions with the appropriate set of ion mass-to-charge ratios for the peptide under study were combined. Further purity control was then performed.

After confirming chromatographic purity of at least 95%, the peptide solutions were packed into sterile vials with a screw cap. The vial contents were lyophilically dried using a ScanVac Coolsafe 110-4 Pro (LaboGene, Allerød, Denmark), weighed and then hermetically sealed and stored at −20 °C. Lyophilisation parameters: temperature −107 °C at a pressure of 0.001–0.002 mbar.

Prior to the experiment, the peptides were dissolved in Tris-HCl buffer (25 mM, pH 8.8) to reach a concentration of 5120 μg/mL.

### 4.9. Measurement of the Antibacterial Activity of Synthetic Peptides

The antimicrobial activity of the peptides was determined using a dilution assay in microtitre plates [33]. *Bacillus subtilis* 168HT and *Escherichia coli* MG1655 cultures were incubated overnight at 30 °C in MHB medium. These cultures were then diluted at a ratio of 1:50 and allowed to grow at 37 °C until they reached the mid-logarithmic phase. The bacterial suspension was then diluted to a concentration of 10^6^ cells/mL. The peptides were diluted by two-fold serial dilution, with final concentrations ranging from 256 to 0.5 μg/mL. A total of 50 μL of medium was mixed with 50 μL of a microbial suspension at a concentration of 10^6^ cells/mL. Each experiment was performed with positive (melittin) and negative (without peptide, with Tris-HCl 25 mM buffer at pH 8.8) controls for bacterial growth inhibition.

Antimicrobial activity was monitored using a Microplate Reader AMR-100 microplate photometer, with the optical density (OD) value recorded at 600 nm. The MIC was defined as the lowest concentration of the peptide that completely inhibited microbial growth.

An additional experiment was performed to evaluate antimicrobial activity by cell spreading on agar MHB medium. For this purpose, 20 µL of the bacterial suspension at a concentration of 10^6^ cells/mL was mixed with 20 µL of the peptide solution at a concentration of 2560 µg/mL, and the mixture was incubated at 37 °C for 30 min. After incubation, the resulting mixture was diluted 100-fold, after which 100 µL of the diluted sample was seeded onto agar medium dishes. The plates were incubated at 37 °C overnight, after which the number of colonies was determined.

## 5. Conclusions

The *H. medicinalis* genome study revealed new potential antimicrobial peptides belonging to the lumbricin and macin families. The most significant finding was the demonstration of high antibacterial activity against the Gram-negative bacterium *E. coli* by peptides LBrHM1, NrlHM1 and NrlHM2. Upon closer analysis of the LBrHM1 peptide, two fragments with different antimicrobial activity were identified. The N-terminal fragment of LBrHM1 has a unique sequence and exhibits high antimicrobial activity against *E. coli* and *B. subtilis* strains, whereas the second fragment predominantly exhibits activity against *B. subtilis*.

## Figures and Tables

**Figure 1 ijms-26-06903-f001:**
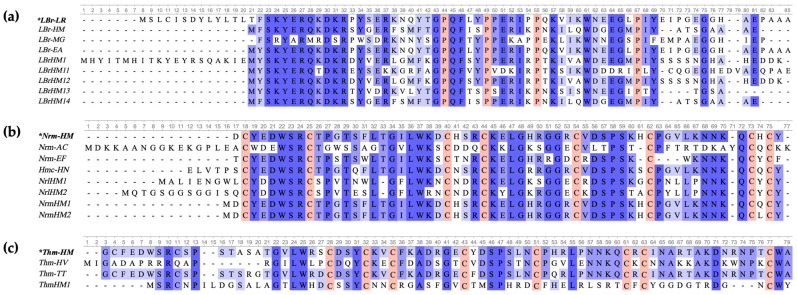
Alignment of amino acid propeptide sequences of *H. medicinalis* (NCBI BioProject PRJNA257563) AMPs of (**a**) lumbricin, (**b**) neuromacin and (**c**) theromacin. LBrHM1, LBrHM11, LBrHM12, LBrHM13, LBrHM14—lumbricin homologues; NrlHM1, NrlHM2, NrmHM1, NrmHM2—neuromacin homologues; ThmHM1—theromacin homologue; LBr-HM—lumbricin (UniProt ID A8V0C6); LBr-LR—lumbricin I (UniProt ID O96447); LBr-MG—lumbricin-PG (UniProt ID A0A288W7G5); LBr-EA—lumbricin (UniProt ID P86929); Nrm-HM—neuromacin (UniProt ID A8V0B3); Nrm-AC—neuromacin-like protein (UniProt ID A5GZY1); Nrm-EF—neuromacin (UniProt ID A0A0M3WPR3); Hmc-HN—hirudomacin (UniProt ID A0A481Y7W7); Thm-HM—theromacin (UniProt ID A8I0L8); Th-HV—theromacin-like protein (UniProt ID A0A0N7Z9P3); Th-TT—theromacin (UniProt ID Q6T6C2). *—Reference sequence. The blue colour gradient reflects the percentage of matching residues at each position: the darker the shade, the higher the degree of conservation and matching of amino acids at a given position. Proline and cysteine residues in patterns are highlighted in red.

**Figure 2 ijms-26-06903-f002:**
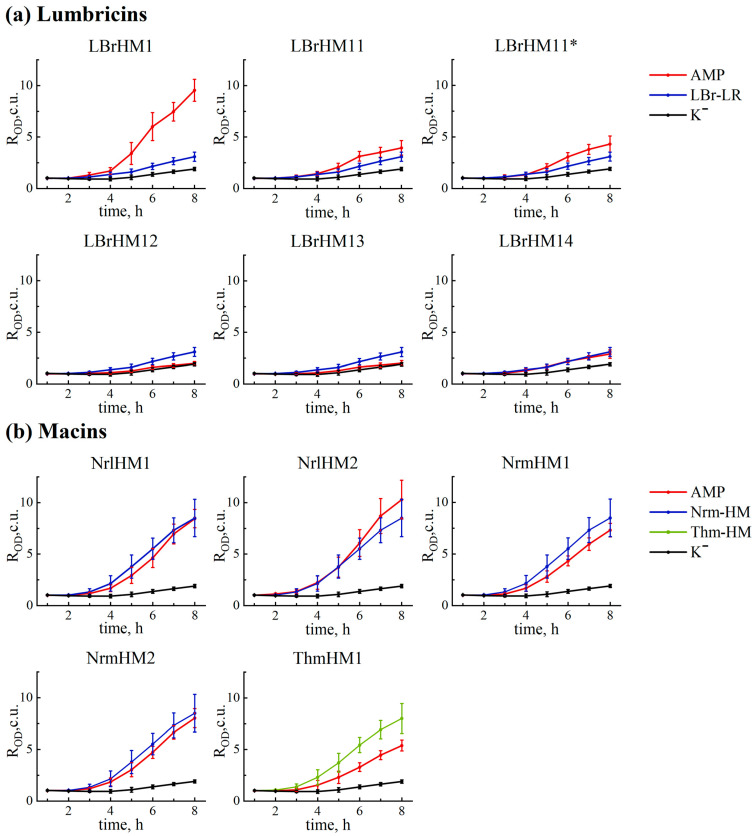
Growth curves of *E. coli* BL21-Gold (DE3) bacterial culture transformed with expression pET-22b(+) plasmids encoding antimicrobial peptides with the PelB signal peptide: (**a**) lumbricins, (**b**) macins. Average optical density ratio for each peptide, calculated as ROD = OD600(-IPTG)/OD600(+IPTG), accompanied by standard deviations. OD600(+IPTG)—optical density value of bacterial culture grown in a medium with 0.1 mM IPTG inducer; OD600(-IPTG)—optical density without transcription inducer. Measurements were taken for sixteen clones every hour. Lines designated as AMP correspond to LBrHM1, LBrHM11, LBrHM12, LBrHM13 and LBrHM14—lumbricin homologues; NrlHM1, NrlHM2, NrmHM1 and NrmHM2—neuromacin homologues; and ThmHM1—theromacin homologue. LBr-LR—lumbricin I (UniProt ID O96447); Nrm-HM—neuromacin (UniProt ID A8V0B3); Thm-HM—theromacin (UniProt ID A8I0L8). K^−^—pET-min.

**Figure 3 ijms-26-06903-f003:**
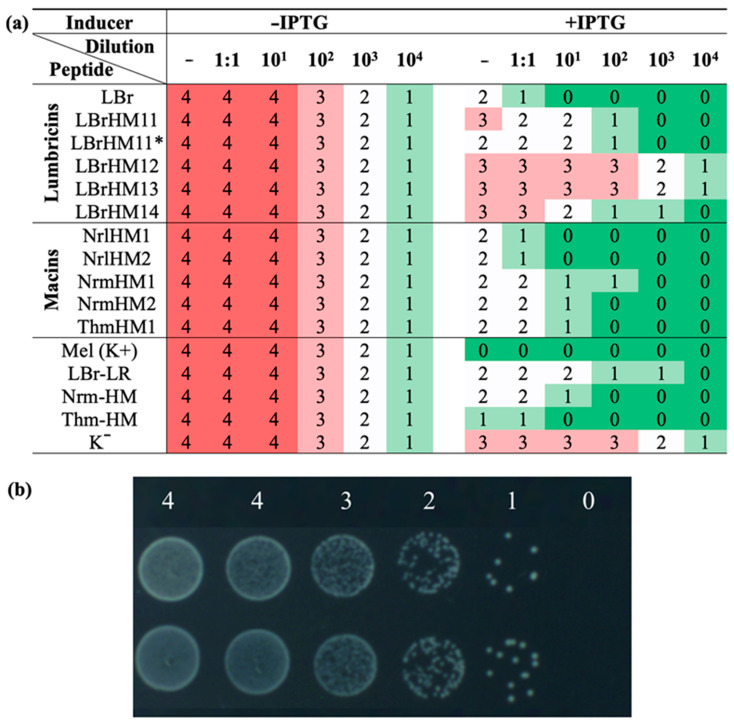
Growth assessment of serial dilution of bacterial cultures expressing antimicrobial peptides on LB agar after 22 h incubation at 37 °C. *E. coli* BL21-Gold (DE3) were transformed with expression plasmids pET-22b(+) encoding antimicrobial peptides: LBrHM1, LBrHM11, LBrHM12, LBrHM13, LBrHM14—lumbricin homologues; NrlHM1, NrlHM2, NrmHM1, NrmHM2—neuromacin homologues; ThmHM1—theromacin homologue; LBr-LR—lumbricin I (UniProt ID O96447); Nrm-HM—neuromacin (UniProt ID A8V0B3); Thm-HM—theromacin (UniProt ID A8I0L8); Mel(K+)—melittin; K^−^—pET-min. (**a**) Digital interpretation of antimicrobial activity analysis on agar. The mode is based on the results of eight experimental replicates and two biological replicates. The values correspond to serial dilutions (no dilution, 1:1, from 10^1^ to 10^4^) on agar medium: −IPTG—without transcription inducer, +IPTG—with the addition of IPTG inducer up to 0.1 mM. Bacterial growth was estimated as follows: (4) a uniform bacterial layer, (3) minor gaps within the bacterial layer, (2) overlapping colonies, (1) single, separately located colonies and (0) no visible colonies. (**b**) Example of drop-plating serial dilution of *E. coli* culture and interpretation of the bacterial growth.

**Figure 4 ijms-26-06903-f004:**
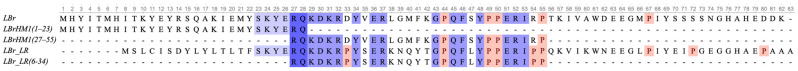
Alignment of lumbricin amino acid sequences. LBrHM1: full-length lumbricin sequence found in the *H. medicinalis* genome; LBrHM1(1–23)—N-terminal fragment; LBrHM1(27–55)—fragment corresponding to the propeptide form of lumbricin I (6–34) [9]; LBr-LR—lumbricin previously identified in *L. rubellus* (UniProt ID O96447); LBr-LR(6–34)—a section from 6 to 34 residues of the mature peptide lumbricin I. The blue colour gradient reflects the percentage of matching residues at each position: the darker the shade, the higher the degree of conservation and matching of amino acids at a given position. Proline residues are highlighted in red.

**Figure 5 ijms-26-06903-f005:**
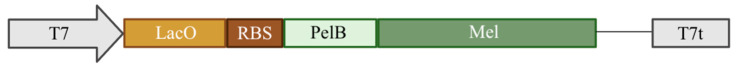
Schematic of the cloning/expression region of the plasmid vector pET-22b(+)—with an insert encoding an antimicrobial peptide. T7—T7 promoter, LacO—lac operator, RBS—ribosome binding site, PelB—signal peptide, Mel—sequence encoding melittin, T7t—T7 terminator.

**Table 1 ijms-26-06903-t001:** Experimental and control plasmid constructs used for expression of potential antimicrobial peptides.

	Peptide Name	Plasmid Name
Experimental	LBrHM1	pET-22b(+)-LBRHM1
	NrlHM1	pET-22b(+)-NRLHM1
	NrlHM2	pET-22b(+)-NRLHM2
	NrmHM1	pET-22b(+)-NRMHM1
	NrmHM2	pET-22b(+)-NRMHM2
	LBrHM11	pET-22b(+)-LBRHM11
	LBrHM11*	pET-22b(+)-LBRHM11*
	LBrHM12	pET-22b(+)-LBRHM12
	LBrHM13	pET-22b(+)-LBRHM13
	LBrHM14	pET-22b(+)-LBRHM14
	ThmHM1	pET-22b(+)-THMHM1
Control	Nrm-HM	pET-22b(+)-NRM-HM
	Thm-HM	pET-22b(+)-THM-HM
	LBr-LR	pET-22b(+)-LBR-LR
	Melittin	pET-22b(+)-mel
	-	pET-min

**Table 2 ijms-26-06903-t002:** Antimicrobial activity of the synthetic peptides LBrHM1, LBrHM1(1–23) and LBrHM1(27–55), as determined by the dilution method.

Peptide	In Liquid Medium		On Agar Medium	
	MIC, µM		CFU	
	*E. coli* MG1655	*B. subtilis* 168HT	*E. coli* MG1655	*B. subtilis 168HT*
LBrHM1	>100	>100	108/323	41/133
LBrHM1_N	>100	70.54	0/0	0/0
LBrHM1_M	>100	71.21	198/239	2/2
Mel (K+)	2.81	1.41	0/0	0/0
25 mM Tris-HCl (pH 8.8)	-	-	521	477
K^−^	-	-	737	713

Note: Results of the assessment of the antimicrobial activity of peptides in two types of medium: liquid and agar. MIC, μM (minimum inhibitory concentration)—the minimum concentration of the peptide (in micromoles) at which bacterial growth is inhibited in liquid medium. CFUs (colony-forming units)—the number of viable bacterial cells capable of forming colonies on agar medium after treatment with peptides. LBrHM1—the complete sequence of lumbricin discovered by us in the genome of *H. medicinalis* (NCBI BioProject PRJNA257563); LBrHM1(1–23)—N-terminal fragment; LBrHM1(27–55)—fragment corresponding to lumbricin I (6–34); Mel (K+)—melittin; 25 mM Tris-HCl (pH 8.8)—with the addition of 25 mM Tris-HCl buffer, not containing peptides; K^−^–pET-min.

**Table 3 ijms-26-06903-t003:** The characteristics of known antimicrobial peptides (AMPs) found in annelid worms.

Family	Abbreviation	Source	Functions	UniProt ID	Ref
Lumbricins	LBr-HM	*Hirudo medicinalis*	Stimulates neuroregeneration	A8V0C6	[10]
	LBr-LR	*Lumbricus rubellus*	Exhibits antimicrobial activity against G+ and G− bacteria, as well as fungi. Does not possess haemolytic activity	O96447	[9]
	LBr-MG	*Metaphire guillelmi*	Exhibits antimicrobial activity against G+ and G− bacteria, as well as fungi. It has very weak haemolytic activity	A0A288W7G5	[20]
	LBr-EA	*Eisenia andrei*	-	P86929	[21]
Macins	Nrm-HM	*Hirudo medicinalis*	Exhibits antimicrobial activity against G+ bacteria, stimulates neuroregeneration	A8V0B3	[10]
	Nrm-AC	*Aplysia californica*	-	A5GZY1	[22]
	Nrm-EF	*Eisenia fetida*	-	A0A0M3WPR3	
	Hmc-HN	*Hirudo nipponia*	Exhibits antimicrobial activity against G+ bacteria	A0A481Y7W7	[12]
	Thm-HM	*Hirudo medicinalis*	Exhibits antimicrobial activity against G+ bacteria, stimulates neuroregeneration	A8I0L8	[11,23]
	Th-HV	*Haementeria vizottoi*	-	A0A0N7Z9P3	[24]
	Th-TT	*Theromyzon tessulatum*	Exhibits antimicrobial activity against G+ bacteria, stimulates neuroregeneration	Q6T6C2	[11]

Note: G+ bacteria—gram-positive bacteria, G− bacteria—gram-negative bacteria.

## Data Availability

The *H. medicinalis* genome is available in the NCBI database under the BioProject accession number PRJNA257563. https://www.ncbi.nlm.nih.gov/bioproject/PRJNA257563/ (accessed on 17 July 2025).

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
