# Peer review of "Identification and Screening of Novel Antimicrobial Peptides from Medicinal Leech via Heterologous Expression in Escherichia coli"

_ijms, 2025, doi:10.3390/ijms26146903_

Round 1
Reviewer 1 Report
Comments and Suggestions for Authors
The authors report a novel screening platform for identifying antimicrobial peptides (AMPs) in the medicinal leech Hirudo medicinalis using heterologous expression in E. coli. They identified several candidate peptides from the lumbricin and macin families, constructed expression vectors, and assessed antimicrobial activity using both optical density measurements and agar-based assays. The presentation of the results in several sections of the manuscript lacks clarity, making it difficult for readers to follow the experimental outcomes and interpret the key findings. The manuscript presents comparative data on the antimicrobial activity of various AMP candidates, including against known positive controls. However, the comparisons are reported descriptively without any accompanying statistical analysis. Please see the detailed comments below.
- The authors have presented the bacterial growth kinetics of coli strains expressing different antimicrobial peptides in Table 2. While the tabular format is informative, it is not the most intuitive for visualizing trends and comparing growth patterns across samples. I strongly recommend supplementing this data with a line graph to provide a clearer, more accessible visualization of the bacterial growth curves. In addition, the current presentation lacks statistical analysis. I suggest including measures of central tendency and variation (e.g., mean ± standard deviation) for each time point based on biological replicates. Applying appropriate statistical tests (such as t-tests or ANOVA) to assess the significance of growth differences between peptide-expressing strains and controls.
- Table 3 presents results that are essentially similar to those shown in Table 2, both reflecting the antimicrobial activity of the peptides using different assay formats. However, presenting both sets of data in tabular form makes it difficult to discern meaningful differences between samples at a glance. It would be better to remove Table 3 and Figure 2 from the manuscript. Instead, replace them with a single composite figure showing representative images of the drop-seeding assay for all tested peptide-expressing strains (with and without IPTG induction). This would allow readers to directly observe the differences in colony formation and antimicrobial effects across the peptide panel, providing a much more intuitive and compelling visualization of the experimental outcomes.
- In Figure 1, the authors present sequence alignments of lumbricin, neuromacin, and theromacin peptides from various species. However, the results section lacks a clear description or interpretation of the comparative analysis shown in these alignments. Additionally, it is not explicitly stated which sequence is used as the reference in each alignment group, making it difficult to assess the degree of conservation or divergence of the newly identified peptides.
- The description of sequence similarities in lines 99-102 is unclear and may lead to confusion. Specifically, it is ambiguous whether the 98% similarity refers to the identity between NrmHM1 and NrmHM2 or to their similarity with a reference neuromacin sequence. Similarly, the comparison basis for the 68% and 64% similarities of NrlHM1 and NrlHM2 is not clearly stated-whether they are being compared to Nrm-HM, to each other, or to another sequence. The same applies to the 36% similarity reported for ThmHM1; the reference sequence used for this comparison should be explicitly mentioned.
- In Section 2.3, the authors evaluate the antimicrobial activity of various AMP candidates and include melittin, LBr-LR, Nrm-HM, and Thm-HM as positive controls for comparison. However, while the experimental design is appropriate, the results do not include any analysis or discussion of how the tested peptides compare to these controls in terms of antimicrobial efficacy.
Author Response
On behalf of all the authors, we would like to sincerely thank you for your comments and suggestions.

Reviewer 2 Report
Comments and Suggestions for Authors
Review- 3733400
Journal: International Journal of Molecular Science
Title: Identification and screening of novel antimicrobial peptides from medicinal leech via heterologous expression in Escherichia coli
This is an excellent piece of work where the authors have demonstrated an innovative screening strategy to identify and characterize antimicrobial peptides (AMPs), which can be the last resort to combat drug-resistance in bacterial pathogens. Authors have identified potential bioactive peptides after data mining of the genome of medicinal leeches and screen those peptides through an innovative screening method. They have cloned these peptides into an expression plasmid under the control of an inducible promoter. After transformation of these plasmid into E. coli, cells were grown under induced and uninduced condition to compare the growth profile. Cells with active peptide show a retardation in growth as compared to uninduced. Overall, this is a very interesting study, and this model can be implemented to screen a AMPs from many natural sources.
Author Response
On behalf of all the authors, we appreciate the positive feedback on our manuscript.
Round 2
Reviewer 1 Report
Comments and Suggestions for Authors
The revised manuscript has significantly improved and now clearly resolves the concerns raised.